# Diagnostic Performance of *Plasmodium falciparum* Histidine-Rich Protein-2 Antigen-Specific Rapid Diagnostic Test in Children at the Peripheral Health Care Level in Nanoro (Burkina Faso)

**DOI:** 10.3390/tropicalmed7120440

**Published:** 2022-12-15

**Authors:** Massa dit Achille Bonko, Marc Christian Tahita, Francois Kiemde, Palpouguini Lompo, Petra F. Mens, Halidou Tinto, Henk. D. F. H. Schallig

**Affiliations:** 1Institut de Recherche en Sciences de la Santé-Direction Régionale du Centre-Ouest/Unité de Recherche Clinique de Nanoro, (IRSS-DRCO/URCN), Nanoro 18, Burkina Faso; 2Laboratory for Experimental Parasitology, Amsterdam University Medical Centers, Department of Medical Microbiology and Infection Prevention, Academic Medical Center at the University of Amsterdam, 1105 AZ Amsterdam, The Netherlands; 3Infectious Diseases Programme, Amsterdam Institute for Infection and Immunity, Amsterdam University Medical Centers, 1105 AZ Amsterdam, The Netherlands

**Keywords:** malaria, diagnostics, Burkina Faso, febrile diseases

## Abstract

(1) Background: Malaria control has strongly benefited from the implementation of rapid diagnostic tests (RDTs). The malaria RDTs used in Burkina Faso, as per the recommendation of the National Malaria Control Program, are based on the detection of histidine-rich protein-2 (*Pf*HRP2) specific to *Plasmodium falciparum*, which is the principal plasmodial species causing malaria in Burkina Faso. However, there is increasing concern about the diagnostic performance of these RDTs in field situations, and so constant monitoring of their accuracy is warranted. (2) Methods: A prospective study was performed in the health district of Nanoro, where 391 febrile children under 5 years with an axillary temperature ≥37.5 °C presenting at participating health facilities were subjected to testing for malaria. The HRP2-based RDT and expert microscopy were used to determine the diagnostic performance of the former. Retrospectively, the correctness of the antimalaria prescriptions was reviewed. (3) Results: Taking expert malaria microscopy as the gold standard, the sensitivity of the employed RDT was 98.5% and the specificity 40.5%, with a moderate agreement between the RDT testing and microscopy. In total, 21.7% of cases received an inappropriate antimalarial treatment based on a retrospective assessment with expert microscopy results. (4) Conclusion: Malaria remains one of the principal causes of febrile illness in Burkina Faso. Testing with HRP2-based RDTs is inaccurate, in particular, due to the low specificity, which results in an over-prescription of antimalarials, with emerging antimalarial drug resistance as an important risk and many children not being treated for potential other causes of fever.

## 1. Introduction

Recent malaria control efforts have resulted in a significant reduction of *Plasmodium falciparum* malaria cases, and associated mortality and morbidity [1]. A global reduction in malaria cases was reported from 2000 to 2019, i.e., 241 million clinical cases in 2000 and 227 million in 2019 [1]. However, these incidence and mortality rates have increased in 2020 with the COVID-19 pandemic that disrupted malaria-control services [1].

The case-detection using malaria rapid diagnostic tests (RDTs) in the test-and-treat strategy advocated by the World Health Organization (WHO) is one of the cornerstones of this success [1]. The majority of malaria RDTs are based on the detection of histidine-rich protein-2 (*Pf*HRP2), which is specific to *Plasmodium falciparum*, or the *Plasmodium*-specific parasite lactate dehydrogenase (*p*LDH), either alone or in combination with *Pf*HRP2. Malaria RDTs are easy to implement at the point-of-care, provide results rapidly, and require no specific training and equipment [2]. However, the performance of these RDTs, and consequently the effectiveness of malaria-control programs, is more and more challenged with increasing numbers of field studies reporting the reduced specificity of the *Pf*HRP2-based RDTs [3,4,5,6]. This leads to over-prescription of antimalarials by health care workers [5,7,8,9].

Alternative diagnostics for malaria include microscopy and molecular-biology-based tests. The latter are considered to be the most sensitive, but infrastructural requirements, training, and costs hamper their large-scale implementation in resource-limited settings [10,11]. Therefore, microscopy detecting of *Plasmodium* parasites in Giemsa-stained thick- or thin-blood slides still remains the gold standard for malaria diagnosis [10,12], even though its sensitivity and specificity depends on the quality of the blood films, the microscopes, and the training of the microscopists.

Burkina Faso, amongst the poorest countries in the world, has a high burden of malaria. According to the World Malaria Report 2021, the whole population of the country is at risk of contracting *Plasmodium falciparum* malaria, and the number of clinically suspected malaria cases was estimated at almost 15 million, with 4000 reported malaria deaths occurring in 2020 [1]. Control efforts in the country involve the confirmation of all malaria cases by RDTs or microscopy before any treatment, and a free-of-charge diagnosis in the public sector. As *P. falciparum* is the predominant malaria species in the country, the National Malaria Control Program (NMCP) has implemented to use of HRP2 RDTs. However, previous studies have demonstrated that its performance in the diagnosis of malaria is reduced [3,4,5,6]. This alarming situation calls for a constant and close monitoring of the performance of these RDTs in the routine setting in order to detect any issue in a timely way [13,14,15,16]. Therefore, the objective of the present study is to assess the performance of the malaria HRP2 RDT recommended by the Burkinabe NMCP at peripheral level compared with expert microscopy at a central reference laboratory.

## 2. Materials and Methods

### 2.1. Study Design and Population

The study was performed between September 2018 and August 2019 in the health district of Nanoro, which is located at approximately 85 km from Ouagadougou, the capital city of Burkina Faso. Children under 5 years presenting with a current axillary temperature ≥37.5 °C at one of the participating health facilities (Nazoanga, Pella, Soaw and Temnaoré) and at the Pediatric ward of “Centre Médical avec antenne chirurgicale (CMA, which is a reference hospital) were recruited for the study after obtaining informed consent from a parent or legal guardian. All participating children were tested for malaria using an *Pf*HRP2 RDT (see below) and managed following NMCP guidelines [17].

The study was approved by the National Ethical Committee “Comité d’Ethique pour la Recherche en Santé” (Deliberation No. 2014-11-130). The study was also approved by the health district authorities and community leaders in different villages before implementation.

### 2.2. Study Procedures

At enrolment, some basic demographic and clinical data were collected from each participant. A small blood sample was collected by finger prick for malaria RDT testing (SD BIOLINE Malaria Ag *P.f*; Standard Diagnostics, Hagal-Dong, Korea) and for microscopy slides preparation. The result of the malaria RDT testing in the health facilities was recorded on a case record form (CRF) as well as the treatment provided for clinical management.

The malaria blood slide was fixed with methanol and stained with 3% Giemsa solution (pH 7.2) for identification and quantification of asexual *P. falciparum* and other *Plasmodium* species. Slide reading was performed by expert microscopists at the Clinical Research Unit of Nanoro (CRUN), who were blinded from the results of the RDT testing.

*P. falciparum* densities were determined by counting the number of asexual parasites per 200 white blood cells and calculated per μL of blood by assuming the number of white blood cells to be at 8000 per μL. Each blood slide was read by two independent expert readers, and in case of discrepancy, a third independent reader was used. Positive microscopy results were recorded as the geometric means of the two readers’ results or the geometric means of the two geometrically closest reading in case of a third reading. These results were expressed as asexual parasites per microliter by using the patient’s white blood cell (WBC) count.

### 2.3. Quality Control

To ensure the quality of sample analysis, either in peripheral health centers or in the central laboratory, all diagnostic procedures were conducted in accordance with the standard operating procedures (SOPs) in place at the CRUN. In addition, there was strict adherence to manufacturers’ guidelines with respect to the RDT testing.

For malaria rapid diagnostic testing based on the histidine-rich protein-2 antigen specific for *Plasmodium falciparum*, the internal control line confirmed the correct performance of the test. There is no further quality assurance system put in place by the Burkinabe NMCP to subject malaria RDTs to for quality control checks before their use. Furthermore, it is important to mention that all expert microscopists of the CRUN laboratory are frequently subjected to the External Quality Control Assessment (EQA) program with the National Institute for Communicable Diseases (NICD, South Africa) and only qualified readers were allowed to read slides [18].

### 2.4. Data Analysis

A double-entry system was performed for data in Excel and checked by an independent data manager. The performance of the RDT was evaluated compared to expert microscopy (gold standard) by calculating the sensitivity, the specificity, and the negative and positive predictive value of the RDT using the MedCalc Software Ltd. diagnostic test evaluation calculator, https://www.medcalc.org/calc/diagnostic_test.php (Version 20.013; accessed on 30 September 2021). The agreement between the RDT and expert microscopy was determined by calculating the kappa (k) values with 95% confidence intervals using GraphPad software, https://www.graphpad.com/quickcalcs/ (accessed on 30 September 2021).

## 3. Results

### 3.1. Characteristics of the Study Population

In total, 391 febrile children under the age of 5 were recruited for this study. The recruitment according to the sites was as follows: 115 were included at the Temnaoré health facility, 91 at the Soaw health facility, 82 at the Nazoanga health facility, 68 at the Pediatric ward of the CMA reference hospital, and 35 at the Pella health facility. The basic characteristics of the study population are presented in Table 1.

### 3.2. Diagnostic Testing

Taking expert malaria microscopy as the gold standard, the sensitivity of the employed RDT was 98.5% and the specificity was 40.5% (details presented in Table 2). The agreement between the RDT and expert microscopy was “moderate” (a kappa value of 0.457).

The HRP2-based RDT was positive for 336 cases (85.9%; 336/391). Expert microscopy found 265 (67.8%; 265/391) cases positive for malaria (see Table 1). All expert malaria microscopy positive slides were *P. falciparum* monoinfections. The RDT testing results combined with the results of the expert malaria microscopy results are presented in Table 3. In total, 261 malaria-positive cases by expert malaria microscopy also had a positive HRP2-based RDT. Furthermore, there were 51 febrile children who had a negative RDT test and a negative expert malaria microscopy result as well. There were 75 children who had a positive RDT but that tested negative by expert malaria microscopy. In contrast, there were four cases that were microscopy-positive but reported to be RDT-negative.

### 3.3. Antimalaria Treatment Provided

Antimalarials, mainly artemisinin-based combination therapy (ACT) comprising of either pyronaridine–artesunate (PA) or *artemether*–*lumefantrine* (*AL*) (both ACTs often combined with paracetamol to suppress fever), were provided to 304 (90.5%; 304/336) of the children who were malaria-RDT-positive. However, there were 32 malaria RDT-positive children (9.5%; 32/336) who did not receive an antimalarial (*pyronaridine–artesunate* or *artemether*–*lumefantrine*) but were treated with antibiotics. Furthermore, 45 RDT-negative children (81.8%; 45/55) were not treated with antimalarials, but they were treated with antibiotics (without further assessment of the actual cause of disease). In addition, 10 (18.2%; 10/55) malaria-RDT-negative children were treated with antimalarials. Overall, there were 349 (89.3%; 349/391) “right” treatments on the basis of the RDT results (i.e., antimalarial was provided if the RDT was positive and not if RDT was negative) and 42 (10.7%; 42/391) “wrong” treatments (i.e., antimalarial was not provided if the RDT was positive and was given if the RDT was negative).

If we considered the malaria-expert-microscopy-positive results, it was found that 247 (92.2%; 247/265) of the children who had malaria-expert-microscopy-positive results received the appropriate antimalaria treatment. However, 18 (6.8%; 18/265) other children in this group did not receive antimalarials. In contrast, 67 (53.2%; 67/126) children who were malaria-expert-microscopy-negative did receive an antimalaria treatment based on the RDT result but not indicated by microscopy. Accordingly, 59 (46.8%; 59/126) microscopy-negative-children were not treated with antimalarials.

In hindsight (using microscopy as confirmation of malaria infection), the actual number of right antimalaria treatments provided to children was 306 (78.3%; 306/391) and the number of wrong treatments was 85 (21.7%; 85/391).

## 4. Discussion

Our study aimed to assess the diagnostic performance of a malaria-HRP2-based RDT recommended by the National Malaria Control Program (NMCP) of Burkina Faso in remote health facilities using microscopy result as the gold standard. Our study revealed high sensitivity (98.5%) of this RDT used at the peripheral level, but with low specificity (40.5%). This confirms previous observations that the HRP2-based RDTs, the most widely employed diagnostic test for malaria, have a low specificity, particularly in a high-transmission setting [3,4,5,6]. This limits its deployment for malaria surveillance, which in turn contributes not only to the over-prescription of anti-malaria treatments, but also to the potential emergence of drug resistance [7,19,20]. This is particularly worrying for the artemisinin-combination therapies (ACTs), which are currently the only effective drugs against malaria [19].

Several factors may have contributed to the low specificity of the employed RDT. First of all, it is well known that the HRP2 antigen persists for approximately three to nine weeks in the blood after successful treatment [21,22,23]. Therefore, it is important to record previous antimalaria treatment and take this into account when diagnosing a patient [21,22,24]. Moreover, the transportation and storage conditions might have affected the test performance as on particular days, as temperatures can be very high in the study area, which may have affected the stability and, consequently, the performance of the RDTs [23,25,26]. Improving the transportation and storage conditions using controlled conditions and temperature-monitoring is therefore recommended. In addition, research towards improving the diagnostic performance of the RDTs, for example, by introducing different target antigens or sequential testing with different malaria RDTs, is warranted [27,28].

Expert microscopy, as performed in the context of the current study, detected a considerably lower number of malaria cases in contrast to those detected by the malaria RDTs recommended by the NMCP of Burkina Faso. As a result of this, if the results of malaria microscopy would have been followed, a significantly lower number of children would have been treated for malaria, thereby saving costs and avoiding the development of resistance. Microscopy has always proven to be a valuable tool for the diagnosis of malaria, and it comes with a good sensitivity/specificity and allows for quantification and species identification. However, its implementation depends on a well-trained microscopist and appropriately maintained equipment, as this is carried out at the CRUN laboratory [18]. Training can be supported by external quality control and assessment [29].

Malaria was found to be the principal cause of febrile illness in the current study. Despite many efforts to control the disease, its prevalence remains high in our health district and calls for increased control efforts. The study was conducted just prior to the outbreak of the COVID-19 pandemic, and, consequently, the malaria situation might have been exacerbated, resulting in an increase of the number of deaths caused by malaria over the last two years, as limited available resources have been directed towards fighting the coronavirus outbreak [1]. Therefore, we call for intensified continuous malaria surveillance that would allow systematic collection, analysis, and interpretation of malaria-related data. This will include, amongst others, *Pf*HRP2/3 deletion data (as these have not yet been identified in Burkina Faso [30]), research data on the reduction of the performance of *Pf*HRP2-based RDTs, and the number of malaria cases and deaths in malaria-endemic regions. In this respect, it is also important to increase research efforts that aim to increase the specificity of HRP2-based RDTs or enable the development of alternative malaria RDTs.

A positive finding of the present study was that the health workers tend to adhere well to the results of the RDT testing, and this is in line with other studies [8,31]. Yet, about 10% of the wrong treatments were given to children, whereby the result of the RDT testing was ignored. It is not clear why the health workers decided to overrule the diagnostic test result. A reason might be the fear of missing a potentially deadly but treatable febrile disease and getting a better understanding of the motives to deviate from the diagnostic testing result should be subject to further research [32]. Furthermore, intensive and frequent training of health workers could further improve their adherence to test results [33].

Finally, it should be noted that there was also a proportion of febrile children who remained undiagnosed (i.e., they are not considered to be malaria cases). In many cases, these children were treated with antibiotics irrespective of the actual cause of their fever. Consequently, the management of febrile diseases may be inappropriate and lead to increase morbidity and mortality, which could be avoided. Moreover, this practice will lead to the development of antibiotic resistance, one of the greatest concerns of modern medicine [7]. It is therefore of utmost importance to develop and evaluate additional simple diagnostic tools for febrile disease, which can be readily implemented in resource limited settings [34].

## 5. Conclusions

Malaria remains one of the principal causes of febrile illness in Burkina Faso, mostly in rural areas. A large proportion of febrile cases in these areas are tested with HRP2-based RDTs, and this frequently yields inaccurate results, in particular, due to its low specificity, which results in an over-prescription of antimalarials that might increase the risk of emerging resistant *Plasmodium* species, and in many children not being treated for potential other causes of fever.

## Figures and Tables

**Table 1 tropicalmed-07-00440-t001:** Baseline characteristics of study population.

Characteristics of Febrile Children Recruited	Study Population
Total, N (%)	391 (100)
Gender	
Male, n (%)	204 (52.2)
Female, n (%)	187 (47.8)
Age in months, mean (min–max)	27.2 (1–58)
Axillary temperature, mean in °C (min–max)	38.7 (36.1–41.5)
Malaria screened by RDT-*Pf*HRP2	
Positive cases, n (%)	336 (85.9)
Negative cases, n (%)	55 (14.1)
Malaria screened by expert microscopy	
Positive cases, n (%)	265 (67.8)
Negative cases, n (%)	126 (32.2)
*Plasmodium* parasites/µL blood, mean (min–max)	100.106 (88–876.250)

N: total number; n: subtotal number; %: percentage; min: minimum; max: maximum; RDT: rapid diagnostic test; *Pf*HRP2: *Plasmodium falciparum* histidine-rich protein-2.

**Table 2 tropicalmed-07-00440-t002:** Diagnostic accuracy of the *Pf*HRP2-based RDT performed at health facilities compared with expert malaria microscopy as the gold standard performed by trained certified microscopists in the reference laboratory.

Accuracy Parameter	Value % (n/N)	Confidence Intervals (95% CI)
Sensitivity	98.5 (261/265)	96.2–99.6
Specificity	40.5 (51/126)	31.8–49.6
Positive predictive value	77.7 (261/336)	75.1–80.1
Negative predictive value	92.7 (51/55)	82.5–97.2

N: total number; n: subtotal number; CI: confidence interval.

**Table 3 tropicalmed-07-00440-t003:** Antimicrobial treatment provided at the recruitment sites combined with malaria diagnostic test results using RDTs and malaria expert microscopy.

Antimicrobial(s) Provided	mRDT-*Pf*HRP2 (N = 391)		Malaria Expert Microscopy (N = 391)		Positive mRDT-*Pf*HRP2 (N = 336)		Negative mRDT-*Pf*HRP2 (N = 55)	
	Positive (N = 336)	Negative (N = 55)	Positive (N = 265)	Negative (N = 126)	Positive Malaria Microscopy (n = 261)	Negative Malaria Microscopy (n = 75)	Positive Malaria Microscopy (n = 4)	Negative Malaria Microscopy (n = 51)
Antimalarial *, n (%)	304 (90.5)	10 (18.2)	247 (93.2)	67 (53.2)	245 (93.9)	59 (78.7)	2 (50.0)	8 (15.7)
Antibiotic, n (%)	108 (32.1)	34 (61.8)	71 (26.8)	71 (56.3)	69 (26.4)	38 (50.7)	2 (50.0)	32 (62.7)

*: antimalarials, mainly pyronaridine-artesunate or artemether-lumefantrine; N: total number; n: sub-total number; %: percentage; mRDT: malaria rapid diagnostic test; *Pf*HRP2: *Plasmodium falciparum* histidine-rich proteins 2.

## Data Availability

All relevant data are within the paper. Anonymized data can be made available upon reasonable request.

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
