# Peer review of "Diagnostic Performance of Plasmodium falciparum Histidine-Rich Protein-2 Antigen-Specific Rapid Diagnostic Test in Children at the Peripheral Health Care Level in Nanoro (Burkina Faso)"

_tropicalmed, 2022, doi:10.3390/tropicalmed7120440_

Round 1

Reviewer 1 Report

Thisistudy contains interesting data about comparison of RDT and microscopy for malaria diagnosis. However there are some major issues with how it has been presented.

1.The title is inappropriate for what is in the paper.  Regarding whether malaira is the principal cause of febrile disease: 1) we are not told how the children were selected for study. 2) other diagnoses (perhaps in addition to malaria ) were not described.  This paper is not about this topic primarily. Regarding whether the RDT is 'increasingly unreliable' 1) this cannot be judged from a cross-sectional study. 2) The study does not show any evidence of HRP deletion which is the main issue for RDTs. In fact microscopy seems to be less sensitive. Title must be changed.

2. The discrepancy between RDT and microscopy is much higher than expected even if antigen persists after treatment. It casts doubt on the accuracy of the microscopy. Or whether the RDT are being read correctly.  

3. It does not say whether the microscopists were blinded to the RDT results

4. To me it would be better to present the diagnostics results first and then go to the treatment results, which are valuable. This is the order in the text, but tables 2 and 3 seem in wrong order. 

5. Conclusion - I would not say that the HRP tests make the process 'cumbersome'. They streamline the process and time to treatment.  But they might be 'inaccurate' (too many false positives).  

To me the results suggest the possibility that the participants (or some of them) may have persisting malaria antigen, but that they may be presenting for an alternative febrile illness. 

Paper needs to be simplified to present the results and not overstate their importance and discuss alternative possibilities.  Discussion needs to refer to other comparative studies of RDT/microscopy.  Remove the reference to 'increasingly unreliable' when you only have one time point. 

Author Response

Dear Editor,

We would like to thank the reviewers for the assessment of our manuscript and their constructive comments. You will find below a point-to-to point rebuttal to the comments made by the two reviewers

Response to Reviewer 1 comments

This study contains interesting data about comparison of RDT and microscopy for malaria diagnosis. However, there are some major issues with how it has been presented.

Point 1: The title is inappropriate for what is in the paper. Regarding whether malaria is the principal cause of febrile disease: 1) we are not told how the children were selected for study. 2) other diagnoses (perhaps in addition to malaria) were not described. This paper is not about this topic primarily. Regarding whether the RDT is 'increasingly unreliable' 1) this cannot be judged from a cross-sectional study. 2) The study does not show any evidence of HRP deletion which is the main issue for RDTs. In fact microscopy seems to be less sensitive. Title must be changed.

Response 1:

We would like to thank the reviewer for the assessment and the suggestions made to improve the manuscript.

To accommodate the reviewer’s comment about the title, we have modified the title as follows: “Malaria diagnosis using a rapid diagnostic test based on histidine-rich protein-2 antigen-specific for Plasmodium falciparum employed at the peripheral health care level in young children in Nanoro (Burkina Faso).”

1) Regarding the recruitment of children, we have described this in the study design and population: children under 5 years who had axillary temperatures higher or equal to 37.5o C were recruited (see lines 98-102).

2) The reviewer is correct that we did not study HRP-2 deletions. So far, there is no evidence in Burkina Faso that these deletions occur (see: Molina-de la Fuente, I., Pastor, A., Herrador, Z. et al. Impact of Plasmodium falciparum pfhrp2 and pfhrp3 gene deletions on malaria control worldwide: a systematic review and meta-analysis. Malar J 20, 276 (2021). https://doi.org/10.1186/s12936-021-03812-0). The main issue in our setting is persistence of HRP-2 antigen. However, we have now included a remark in the discussion session that future studies in our region could also include further research towards HRP2/3 deletions.

Point 2: The discrepancy between RDT and microscopy is much higher than expected even if antigen persists after treatment. It casts doubt on the accuracy of the microscopy. Or whether the RDT are being read correctly.

Response 2:  

Our microscopists are subjected to an extensive training programme and microscopy is subjected to vigorous quality control (see ref. 19: Tinto et al Malar J. 2014;13:113 ). Health staff in the local health centres are trained in performing RDTs and tend to do this according to acceptable standards (see Ruizendaal et al Am J Trop Med Hyg. 2017 Oct;97(4):1190-1197). We do not think that there needs to be doubt on the performance of microscopy or RDT testing.

Point 3: It does not say whether the microscopists were blinded to the RDT results.

Response 3:

In our study, the expert microscopists have been blinded to the RDT results. This is now specifically mentioned in the manuscript. (see lines 117-118).

Point 4: To me it would be better to present the diagnostics results first and then go to the treatment results, which are valuable. This is the order in the text, but tables 2 and 3 seem in wrong order.

Response 4:

We value the reviewer's suggestion and we have put these tables in the right order i.e., table 3 has become table 2 and former table 2 has now become table 3 (see in 3.2. Diagnostic testing part).

Point 5: I would not say that the HRP tests make the process 'cumbersome'. They streamline the process and time to treatment. But they might be 'inaccurate' (too many false positives).

Response 5:

We agree with this comment and have modified the text in the abstract (line 52) and the conclusion (line 271) accordingly.

Point 6: To me the results suggest the possibility that the participants (or some of them) may have persisting malaria antigen, but that they may be presenting for an alternative febrile illness.

Paper needs to be simplified to present the results and not overstate their importance and discuss alternative possibilities. Discussion needs to refer to other comparative studies of RDT/microscopy. Remove the reference to 'increasingly unreliable' when you only have one time point.

Response 6:

We agree with the reviewer that persisting antigen is a major concern (as mentioned in lines 216-217 of the discussion. Also, the fact that other causes of febrile illness play a role has been addressed (see lines 259-261 of the discussion). We think that we have thus adequately addressed the points raised by the reviewer. References 21- - 28 have been used in the discussion to further set the context of the work.  We have removed the reference to reference to 'increasingly unreliable'

We hope that we have now sufficiently addressed the issues raised by the three reviewers and that you will consider our manuscript for publication in your journal.

Yours sincerely,

On behalf of all co-authors

Massa dit Achille Bonko.

Reviewer 2 Report

This current study presents the unreliability of the Rapid Diagnostic Test (RDT) with low specificity, discusses the major consequence of RDT which is an over prescription of antimalarials increasing the risk of antimalarial drug resistance. However there are many gaps in the study which need to be addressed.

  1. Section 1. Introduction, the first line mentions the significant reduction in the numbers of the P. falciparum malaria cases, kindly include the numbers from the cited reference. It is advised to mention the global toll in terms of malaria cases and deaths, worldwide and also, in Burkina Faso region.

  1. The title of the study is not appropriate as the objective of this study as mentioned in the manuscript was to assess the performance of the malaria histidine rich protein 2 (HRP2) RDT recommended by Burkina NMCP at peripheral level compared with expert microscopy at a cereal reference laboratory (Line 87-88). The study is done to assess the diagnostic performance of RDT, which should be precisely mentioned in the title, instead of mentioning about the principal cause of febrile disease in young children. However this information could be included in the introduction section of the manuscript.
  2. The article has presented the case study w.r.t. P. falciparum malaria cases and the diagnostics, which should also be clear in the title of the study.

  1. Lines 65-67 mention about the decreased specificity of PfHRP2 based RDTs. It is a well known fact that deletions in parasite’s PfHRP2/3 genes renders parasites undetectable by RDTs, that are based on HRP2. WHO recommends that countries with reports of deletion of PfHRP2/3 deletions should conduct representative baseline surveys among suspected malaria cases to determine whether the prevalence of PfHRP2/3 deletions causing false negative RDT results has reached a threshold for a change in RDT; >5% deletions causing false negative RDT results (WHO MALARIA REPORT 2021). Kindly Include this information in your introduction and the study design, whether you took that fact into consideration as already recommended by WHO. Has it been published about any reports of deletions of the genes in Burkina Faso region., since there had been many reports which state the deletion PfHRP2 deletions in almost 37 countries (Malaria Threat Maps).
  2. Lines 191-192 mentions again about the low specificity of HRP2 based RDT but doesn’t mention about the one of the causes of low specificity which is deletion of the gene PfHRP2. 
  3. Lines 77-78, 15 million clinical cases and deaths are mentioned but it is not clear if those numbers are worldwide or w.r.t. Burkina Faso. Please paraphrase that information with suitable references. 
  4. Please include if it has been reported or not whether PfHRP2 deletion has been detected in Burkina Faso, along with the intensified malaria surveillance (lines 225-7). Discuss on the suggestive malaria surveillance programs which could improve the specificity of RDTs or alternatives.
  5. Results section 3.3, lines 168-169, what was the rationale behind including the 45 RDT negative children with fever, not treated with antimalarials in this malaria study, similarly, 32 RDT positive children were not given antimalarial treatment but antibiotics.
  6. Line 235 is unclear, please change the words like this with the exact context.
  7. Kindly rewrite the lines 239-40 (improve grammar).

Author Response

Dear Editor,

We would like to thank the reviewers for the assessment of our manuscript and their constructive comments. You will find below a point-to-to point rebuttal to the comments made by the two reviewers

Response to Reviewer 2 comments

Point 1: This current study presents the unreliability Diagnostic Test (RDT) with low specificity, discusses the major of the Rapid consequence of RDT which is an over prescription of antimalarials increasing the risk of antimalarial drug resistance. However there are many gaps in the study which need to be addressed.

Response 1:

We thank the reviewer for the assessment and have addressed the comments below.

Point 2:

1. Section 1. Introduction, the first line mentions the significant reduction in the numbers of the P. falciparum malaria cases, kindly include the numbers from the cited reference. It is advised to mention the global toll in terms of malaria cases and deaths, worldwide and also, in Burkina Faso region.

Response 2:

We have now included the global toll in terms of malaria cases and death, worldwide and in the Burkina Faso region in the introduction (see lines 59-60 and lines 81 - 84).

Point 3:

2. The title of the study is not appropriate as the objective of this study as mentioned in the manuscript was to assess the performance of the malaria histidine rich protein 2 (HRP2) RDT recommended by Burkina NMCP at peripheral level compared with expert microscopy at a cereal reference laboratory (Line 87-88). The study is done to assess the diagnostic performance of RDT, which should be precisely mentioned in the title, instead of mentioning about the principal cause of febrile disease in young children. However, this information could be included in the introduction section of the manuscript.

Response 3:

We agree with the reviewer's comments and suggestions regarding the paper title and we have modified the title accordingly.

Point 4:

3. The article has presented the case study w.r.t. P. falciparum malaria cases and the diagnostics, which should also be clear in the title of the study.

Response 4:

We have now changed the title accordingly.

Point 5:

4. Lines 65-67 mention about the decreased specificity of PfHRP2 based RDTs. It is a well known fact that deletions in parasite’s PfHRP2/3 genes renders parasites undetectable by RDTs, that are based on HRP2. WHO recommends that countries with reports of deletion of PfHRP2/3 deletions should conduct representative baseline surveys among suspected malaria cases to determine whether the prevalence of PfHRP2/3 deletions causing false negative RDT results has reached a threshold for a change in RDT; >5% deletions causing false negative RDT results (WHO MALARIA REPORT 2021). Kindly Include this information in your introduction and the study design, whether you took that fact into consideration as already recommended by WHO. Has it been published about any reports of deletions of the genes in Burkina Faso region., since there had been many reports which state the deletion PfHRP2 deletions in almost 37 countries (Malaria Threat Maps).

Response 5:

We agree with the reviewer that these types of deletions are important to monitor. However, we did not do this in the present study, but do mention it as an important factor in the discussion (see lines 245-247) and we suggest considering the occurrence of these mutations in future studies. It is noteworthy that according to a very recent systematic review and meta-analysis, thePfHRP2/3 deletions have not (yet) been found in Burkina Faso. (Reference: Molina-de la Fuente, I., Pastor, A., Herrador, Z. et al. Impact of Plasmodium falciparum pfhrp2 and pfhrp3 gene deletions on malaria control worldwide: a systematic review and meta-analysis. Malar J 20, 276 (2021). https://doi.org/10.1186/s12936-021-03812-0)

Point 6:

5. Lines 191-192 mentions again about the low specificity of HRP2 based RDT but doesn’t mention about the one of the causes of low specificity which is deletion of the gene PfHRP2.

Response 6: see our rebuttal above. At this moment there is no reason to believe that this is affecting the diagnostic performance in our setting.

Point 7:

6. Lines 77-78, 15 million clinical cases and deaths are mentioned but it is not clear if those numbers are worldwide or w.r.t. Burkina Faso. Please paraphrase that information with suitable references.

Response 7: This observation is relevant; therefore, we have clarified the information in the manuscript (lines 81-84) and clearly defined that those numbers are for Burkina Faso (Please on pages 236 [ANNEXE 5-H. REPORTED MALARIA CASES BY METHOD OF CONFIRMATION, 2010-2020] and 260 [ANNEXE 5-J. REPORTED MALARIA DEATHS, 2010-2020] of WHO. World malaria report 2021(reference 1 of the paper).

Point 8:

7. Please include if it has been reported or not whether PfHRP2 deletion has been detected in Burkina Faso, along with the intensified malaria surveillance (lines 225-7). Discuss on the suggestive malaria surveillance programs which could improve the specificity of RDTs or alternatives.

Response 8:

We have mentioned that Pf-HRP2 deletion has not been detected in Burkina Faso yet and we discussed suggestive malaria surveillance programs to improve the performance of RDTs or alternatives (lines 243-249).

Point 9:

8. Results section 3.3, lines 168-169, what was the rationale behind including the 45 RDT negative children with fever, not treated with antimalarials in this malaria study, similarly, 32 RDT positive children were not given antimalarial treatment but antibiotics.

Response 9:

The criteria for inclusion were age (children under 5 years) and axillary temperature (³37.5 °C) (see study design and population section lines106-110). Therefore, in our analysis, we have taken into account these children to assess the effect of the RDT results on the antimicrobial prescriptions (anti-malarial and antibiotics). Thus, these children were not included on the basis of RDT results but on the basis of their febrile symptoms.

Point 10:

9. Line 235 is unclear, please change the words like this with the exact context.

Response 10:

We have rewritten this line (see lines 255).

Point 11:

10. Kindly rewrite the lines 239-40 (improve grammar).

Response 11:

We have rewritten these lines 254-256 (see lines 254-256).

We hope that we have now sufficiently addressed the issues raised by the three reviewers and that you will consider our manuscript for publication in your journal.

Yours sincerely,

On behalf of all co-authors

Massa dit Achille Bonko.

Reviewer 3 Report

Overall Comments :

The purpose of the study is to compare the diagnostic performance of RDTs used by the NMCP in the study area with microscopy (Gold standard). It’s not clear whether only monovalent RDTs(HRP2) are used/recommended by the NMCP. If that is the case how is the diagnosis of other malaria parasites prevalent in the area done? What is the proportion of Pf, Pv, and other malaria parasites in the study area? Clarity is needed on the diagnosis policy, and type of RDTs used by the Programme. The introduction doesn’t sound convincing. If the situation is already alarming, what cognizance has been taken by the NMCP? Does a quality assurance mechanism exist or not? There is too much vagueness in the entire write-up. The English langue also requires major revisions for clarity. The discussion is totally off track.

Specific Comments :

Title: It is advised to change the title in line with the objective of the study, avoiding repetition of words (as in the present title). The title should be simple and straightforward.

Abstract :

1.      What do the authors mean by “ a significant proportion of the RDTs is based on the detection of histidine-rich protein-2 (PfHRP2), which is specific to Plasmodium falciparum, including those used in the context of the National Malaria Control Program in Burkina Faso? This needs to be clarified and quantified. What are the other RDTs used?

2.      While using the term PfHRP2-based RDT, authors need to specify whether they are monovalent or bivalent RDTs. What is used and recommended by the National Program?

3.      It has been repeatedly claimed that testing with RDTs is cumbersome, while the fact is that RDTs are field as well as user-Introduction:

 1.      The diagnostic performance of different RDT products varies. While talking about diagnostic performance, it is important to mention what RDT quality assurance mechanisms are in place in the same way as the authors have mentioned about microscopy, given the fact that RDT as well as microscopy require quality assurance. 

2. The background should cover the parasite proportion, recommended RDT types as well as the quality assurance policy in the study area.

 Methodology :

1.      What is the reason and logic of recording microscopy positive results as the geometric means? Pl explain.

2.      What were the inclusion and exclusion criteria of the study participants? How many health facilities were included in the study and what was the no. of study participants in each health facility?

3.      Was there a specific reason for recruiting only < 5 yrs. Children in the study?

   Results & discussion :

1.      The study appears to be biased. How could 100%  microscopy results be mono infections with Pf?

2.      Why did the authors not record previous anti-malaria treatment and take this into account when including a patient in the study, given the fact that antigen persistence is expected for at least three weeks after treatment?

3.      So many possibilities for low specificity have been enumerated. Since this was a controlled study specifically designed to assess the diagnostic performance of a particular type of RDT.

In such a study the RDTs used are not expected to already deteriorate due to various reasons.

4.      Overdiagnosis of PF infections due to the persistence of antigen and repeated testing with RDTs is well known, but no program is expected to procure and use RDTs with specificity as low as 40 %.

5.      What is the message of the RDT policy of the country? How do the authors perceive this study to be useful for the Program?

     References:

References have not been appropriately quoted in many instances. This needs to be checked. 

Author Response

Dear Editor,

We would like to thank the reviewers for the assessment of our manuscript and their constructive comments. You will find below a point-to-to point rebuttal to the comments made by the two reviewers

Response to Reviewer 3 comments

Point 1: The purpose of the study is to compare the diagnostic performance of RDTs used by the NMCP in the study area with microscopy (Gold standard). It’s not clear whether only monovalent RDTs(HRP2) are used/recommended by the NMCP. If that is the case how is the diagnosis of other malaria parasites prevalent in the area done? What is the proportion of Pf, Pv, and other malaria parasites in the study area? Clarity is needed on the diagnosis policy, and type of RDTs used by the Programme. The introduction doesn’t sound convincing. If the situation is already alarming, what cognizance has been taken by the NMCP? Does a quality assurance mechanism exist or not? There is too much vagueness in the entire write-up. The English langue also requires major revisions for clarity. The discussion is totally off track.

Response 1:

We thank the reviewer for the assessment of the manuscript. Indeed, monovalent RDTs are recommended by the Burkinabe NMCP as other Plasmodium species, next to P. falciparum, hardly occur in our country,

Thanks to the comments of the three reviewers, we have tried to improve the clarity of the manuscript. English revision was done.

Specific comments:

Point 2: Title: It is advised to change the title in line with the objective of the study, avoiding repetition of words (as in the present title). The title should be simple and straightforward.

Response 2:

We have considered the reviewer's suggestions regarding the paper title and we have now modified the title.

Abstract:

Point 3:

1. What do the authors mean by “ a significant proportion of the RDTs is based on the detection of histidine-rich protein-2 (PfHRP2), which is specific to Plasmodium falciparum, including those used in the context of the National Malaria Control Program in Burkina Faso? This needs to be clarified and quantified. What are the other RDTs used?

Response 3:

Many countries, including Burkina Faso, only use HRP-2-based RDTs for the diagnosis of malaria. We do not have exact numbers further.

Point 4:

2. While using the term PfHRP2-based RDT, authors need to specify whether they are monovalent or bivalent RDTs. What is used and recommended by the National Program?

Response 4:

The NMCP of Burkina Faso recommends the use of PfHRP2-based RDT. This has been stipulated in the manuscript.

Point 5:

3. It has been repeatedly claimed that testing with RDTs is cumbersome, while the fact is that RDTs are field as well as user

Response 5: We have removed the term cumbersome form the manuscript

Introduction:

Point 6:

1. The diagnostic performance of different RDT products varies. While talking about diagnostic performance, it is important to mention what RDT quality assurance mechanisms are in place in the same way as the authors have mentioned about microscopy, given the fact that RDT as well as microscopy require quality assurance.

Response 6:

To accommodate the reviewer’s comment we have now introduced an additional section on QC aspects (section 2.4).

Point 7:

2. The background should cover the parasite proportion, recommended RDT types as well as the quality assurance policy in the study area.

Response 7: We have indicated in the introduction that P. falciparum is the main malaria species in our region and that therefore the NMCP has adopted PF-HRP2 tests. As P. falciparum is the predominant malaria species in the country, the National Malaria Control Program (NMCP) has implemented to use of HRP2 RDTs QC aspects have been addressed in section 2.4.

Methodology:

Point 8:

1. What is the reason and logic of recording microscopy positive results as the geometric means? Pl explain.

Response 8:

The reason and logic for recording/calculating parasite density as geometric means is the fact that the geometric mean is more accurate than the arithmetic mean when there is volatility in the data set.  This is the reason why the geometric mean is often used in biology to calculate certain parameters that do not follow a normal (overdispersion). This is also the case for parasite density.

Point 9:

2. What were the inclusion and exclusion criteria of the study participants? How many health facilities were included in the study and what was the no. of study participants in each health facility?

Response 9:

The inclusion and exclusion criteria have been listed in section 2.1. There were 5 health facilities included in the study (see also section 2.5). The number of participants was 391 febrile children under 5 years of whom 115 were included at Temnaoré health facility, 91 at Soaw health facility, 82 at Nazoanga health facility, 68 at the Paediatric ward of CMA reference hospital, and 35 at Pella health facility.

Point 10:

3. Was there a specific reason for recruiting only < 5 yrs. Children in the study?

Response 10:

This is the major population at risk (next to pregnant women).

Results & discussion:

Point 11:

1. The study appears to be biased. How could 100% microscopy results be mono infections with Pf?

Response 11:

This is the situation in Burkina Faso (at least in our study region), where we do not observe other Plasmodium infections.

Point 12:

2. Why did the authors not record previous antimalaria treatment and take this into account when including a patient in the study, given the fact that antigen persistence is expected for at least three weeks after treatment?

Response 12:

This is indeed a lesson learned from the study and a good suggestion. We will record this in future studies.

Point 13:

3. So many possibilities for low specificity have been enumerated. Since this was a controlled study specifically designed to assess the diagnostic performance of a particular type of RDT.

In such a study the RDTs used are not expected to already deteriorate due to various reasons.

Response 13:

We are not sure what the reviewing is hinting for with this comment. Indeed, there are many reasons for the low specificity and we have tried to address them all. With respect to deterioration, it should be noted that conditions are very harsh in rural Burkina Faso with temperatures often reaching >45 C.

Point 14:

4. Overdiagnosis of PF infections due to the persistence of antigen and repeated testing with RDTs is well known, but no program is expected to procure and use RDTs with specificity as low as 40 %.

Response 14:

We agree and we hope that we get this message through competent authorities. This is an aim of our manuscript.

Point 15:

5. What is the message of the RDT policy of the country? How do the authors perceive this study to be useful for the Program?

Response 15:

We hope to raise awareness of the rather poor performance of the current employed RDT and we also advocate for more intensified surveillance, including monitoring of possible emergence of HRP2/3 resistance.

References:

Point 16: References have not been appropriately quoted in many instances. This needs to be checked.

Response 16:

This has now been done. An additional reference has been added: Molina-de la Fuente, I., Pastor, A., Herrador, Z. et al. Impact of Plasmodium falciparum pfhrp2 and pfhrp3 gene deletions on malaria control worldwide: a systematic review and meta-analysis. Malar J 20, 276 (2021). https://doi.org/10.1186/s12936-021-03812-0

We hope that we have now sufficiently addressed the issues raised by the three reviewers and that you will consider our manuscript for publication in your journal.

Yours sincerely,

On behalf of all co-authors

Massa dit Achille Bonko.

Round 2

Reviewer 1 Report

The manuscript is improved. The title is now more appropriate. However if they are now making the main finding that malaria is the principal cause of febrile illness, they need to confirm how the children were selected for malaria testing.  Either ALL children with fever >37.5 deg C were referred for testing, or a subset who were selected with clinical suspicion of malaria (or to rule out malaria). What about those with fever in last 48 hrs or last week? Were they not tested?  If ALL febrile children, then just say so. If a subset with suspected malaria, how were they selected?  Was it those thought to have malaria on clinical grounds, or where no other cause identified?

The test positivity rate (RDT or slide) is very high, so it suggests that there was some kind of selection process. If not that is OK but it needs to be stated. 

In the abstract conclusion, it is true that overpresciption of antimalarials may be an issue, but also the children are not getting treated for other possible causes of fever. This seems more important to me than potential drug resistance. 

In lines 260 and 262 I am not sure what is meant by 'hit and lead'.   Please rephrase this sentence.

Author Response

Dear Editor,

We would like to thank the reviewers for the assessment of our manuscript and their constructive comments to improve this manuscript. You will find below a point-to-to point rebuttal to the comments made by the two reviewers

Response to Reviewer 1

Point 1: The manuscript is improved. The title is now more appropriate.

Response 1: Thank you for the observation and we are happy that reviewer 1 finds the title now more appropriate.

Point 2: However, if they are now making the main finding that malaria is the principal cause of febrile illness, they need to confirm how the children were selected for malaria testing. Either ALL children with fever >37.5 deg C were referred for testing, or a subset who were selected with clinical suspicion of malaria (or to rule out malaria). What about those with fever in last 48 hrs or last week? Were they not tested? If ALL febrile children, then just say so. If a subset with suspected malaria, how were they selected? Was it those thought to have malaria on clinical rounds, or where no other cause identified?

Response 2: Indeed, all children presenting with a current fever were selected for RDT testing. Thus, no other criteria were applied. We think we have stated this clearly now in the Materials section (see 2.1: Study design and population lines 98-102).

Point 3: The test positivity rate (RDT or slide) is very high, so it suggests that there was some kind of selection process. If not that is OK but it needs to be stated.

Response 3:  There is no specific selection process. As mentioned in the Materials section the only inclusion criterion was presenting with fever (at that particular moment). We think we have adequately mentioned this in the text (see 2.1: Study design and population lines 98-102).

Point 4: In the abstract conclusion, it is true that overpresciption of antimalarials may be an issue, but also the children are not getting treated for other possible causes of fever. This seems more important to me than potential drug resistance.

Response 4: We have now added to the conclusion (both abstract as well as main text): “and many children not being treated for potential other causes of fever.” (see line 53-54).

Point 5: In lines 260 and 262 I am not sure what is meant by 'hit and lead'. Please rephrase this sentence.

Response 5: We have modified this sentence by removing “take a hit” and replacing it with ”be inappropriate” (see lines 265-266).

We hope that we have now sufficiently addressed the issues raised by the three reviewers and that you will consider our manuscript for publication in your journal.

Yours sincerely,

On behalf of all co-authors

Massa dit Achille Bonko.

Reviewer 3 Report

Comments and suggestions to the authors :

The comments and changes in the revised manuscript have been reviewed and the point-wise comments and suggestions are provided below for further consideration by the authors :

S.No.

Review Comments - Version 1(Original Paper)

Review Comments - Version 2

(Revised Paper)

1.       

Title: It is advised to change the title in line with the objective of the study, avoiding repetition of words (as in the present title). The title should be simple and straightforward.

Though the authors have revised the title, it is suggested authors may consider revising it again for clarity as given below :

Diagnostic performance of P. falciparum (Pf) histidine-rich protein 2 (HRP2) antigen-specific rapid diagnostic tests in children at the peripheral level in Nanoro (Burkina Faso)

2.       

Abstract :

i)What do the authors mean by “ a significant proportion of the RDTs is based on the detection of histidine-rich protein-2 (PfHRP2), which is specific to Plasmodium falciparum, including those used in the context of the National Malaria Control Program in Burkina Faso? This needs to be clarified and quantified. What are the other RDTs used?

ii) While using the term PfHRP2-based RDT, authors need to specify whether they are monovalent or bivalent RDTs. What is used and recommended by the National Program?

iii) It has been repeatedly claimed that testing with RDTs is cumbersome, while the fact is that RDTs are field as well as user-friendly. Why

i)                 Though the authors have provided the clarification, appropriate modification in the abstract has not been made ( Lines 39-42). Since the details are not available with authors and the study is specific to  Nanoro in Burkina, it is desirable that the text is context specific and modified  accordingly to include the Pf prevalence in Burkina Faso and the policy  of use of P. falciparum (Pf) histidine-rich protein 2 (HRP2) antigen-specific rapid diagnostic tests as per NMCP recommendations for diagnosis of malaria.

 ii)  – Addressed above

iii)- Addressed by authors in the revised version of the paper.

3.       

Introduction:

i)      The diagnostic performance of different RDT products varies. While talking about diagnostic performance, it is important to mention what RDT quality assurance mechanisms are in place in the same way as the authors have mentioned about microscopy, given the fact that RDT as well as microscopy require quality assurance. 

ii)    The background should cover the parasite proportion, recommended RDT types as well as the quality assurance policy in the study area.

 i)                 Partly addressed.

 For information of the authors , the response expected was regarding the conformation of  RDTs used to WHO product and  lot testing standards. In case this information is available , authors are requested to add it under the Quality Assurance heading.  In case the mechanism does not exist , the changes done by authors can be accepted.

 ii)  Not addressed. However, accepted in view of the general statement in text regarding Pf predominance and author’s response.

Methodology :

i)                 What is the reason and logic of recording microscopy positive results as the geometric means? Pl explain.

ii)               What were the inclusion and exclusion criteria of the study participants? How many health facilities were included in the study and what was the no. of study participants in each health facility,

iii)             Was there a specific reason for recruiting only < 5 yrs. Children in the study?

 i) Addressed by authors and  accepted.

 ii) Addressed partially. The breakup of 391 children has been given in the reply but I couldn’t see it in the revised paper. Pl add this.

iii)             Response provided by authors is accepted.

Results & Discussion :

i)                 The study appears to be biased. How could 100%  microscopy results be mono infections with Pf?ii) Why did the authors not record previous anti-malaria treatment and take this into account when including a patient in the study, given the fact that antigen persistence is expected for at least three weeks after treatment?

iii)             So many possibilities for low specificity have been enumerated. Since this was a controlled study specifically designed to assess the diagnostic performance of a particular type of RDT, in such a study the RDTs used are not expected to already deteriorate due to various reasons.

iv) Over diagnosis of PF infections due to the persistence of antigen and repeated testing with RDTs is well known, but no program is expected to procure and use RDTs with specificity as low as 40 %.

v)               What is the message of the RDT policy of the country? How do the authors perceive this study to be useful for the Program?

i)Addressed; response accepted.

ii)Not addressed. However, since the authors do not have a way of addressing this issue now, the response of authors is accepted.

iii) It seems the authors have not been able to understand the query. This is related to the quality assurance mechanisms of RDTs as per WHO guidelines. The  RDTs procured and used by the NMCP are expected to be subjected to quality checks through a quality assurance system.  Incase such mechanisms are in place ,  it is desirable to add the same under the quality assurance/quality control heading as already explained above. This will add value to the paper.

iv)   

Again, this is related to the quality assurance of RDTs. If the system exists , pl add under relevant heading as already explained above.

 v) Though the authors have responded to this, it is advised that in view of above comments and depending upon the quality assurance mechanisms present in the country under the NMCP, the appropriate information should be provided under the quality assurance heading. Incase, no such mechanisms exist, this should be written clearly and accordingly addressed under the discussion and conclusion also, so that the limitations arising due to quality assurance issues are brought clearly. This would be useful for appropriate policy changes by the NMCP also.  

Author Response

Dear Editor,

We would like to thank the reviewers for the assessment of our manuscript and their constructive comments to improve this manuscript. You will find below a point-to-to point rebuttal to the comments made by the two reviewers

Response to Reviewer 2:

We thank reviewer 2 for providing extensive feedback and putting this in a table format. We have placed our responses in this table too.

S.No.

Review Comments - Version 1(Original Paper)

Review Comments - Version 2

(Revised Paper)

Rebuttal by authors

1.

Title: It is advised to change the title in line with the objective of the study, avoiding repetition of words (as in the present title). The title should be simple and straightforward.

Point 1: Though the authors have revised the title, it is suggested authors may consider revising it again for clarity as given below :

Diagnostic performance of P. falciparum (Pf) histidine-rich protein 2 (HRP2) antigen-specific rapid diagnostic tests in children at the peripheral level in Nanoro (Burkina Faso)

Response 1:

We thank the reviewer for the suggestion and we have now modified the title accordingly. We have left out the abbreviations for clarity. Also, we have used test (and not tests) as it is a single test that was evaluated.

2.

Abstract :

i)What do the authors mean by “ a significant proportion of the RDTs is based on the detection of histidine-rich protein-2 (PfHRP2), which is specific to Plasmodium falciparum, including those used in the context of the National Malaria Control Program in Burkina Faso? This needs to be clarified and quantified. What are the other RDTs used?

ii) While using the term PfHRP2-based RDT, authors need to specify whether they are monovalent or bivalent RDTs. What is used and recommended by the National Program?

iii) It has been repeatedly claimed that testing with RDTs is cumbersome, while the fact is that RDTs are field as well as user-friendly. Why

Point 2: i) Though the authors have provided the clarification, appropriate modification in the abstract has not been made ( Lines 39-42). Since the details are not available with authors and the study is specific to Nanoro in Burkina, it is desirable that the text is context specific and modified accordingly to include the Pf prevalence in Burkina Faso and the policy of use of P. falciparum (Pf) histidine-rich protein 2 (HRP2) antigen-specific rapid diagnostic tests as per NMCP recommendations for diagnosis of malaria.

Point 3: ii) – Addressed above

Point 4: iii)- Addressed by authors in the revised version of the paper.

Response 2:

i) We have now modified the abstract to state that in Burkina Faso only HRP-2-based RDTs are recommended by the NMCP (see lines 38-41).

Response 3:

ii) See comment above

Response 4:

iii) No need to address this further

3.

Introduction:

i) The diagnostic performance of different RDT products varies. While talking about diagnostic performance, it is important to mention what RDT quality assurance mechanisms are in place in the same way as the authors have mentioned about microscopy, given the fact that RDT as well as microscopy require quality assurance.

ii) The background should cover the parasite proportion, recommended RDT types as well as the quality assurance policy in the study area.

Point 5: i) Partly addressed.

For information of the authors , the response expected was regarding the conformation of RDTs used to WHO product and lot testing standards. In case this information is available , authors are requested to add it under the Quality Assurance heading. In case the mechanism does not exist , the changes done by authors can be accepted.

Point 6: ii) Not addressed. However, accepted in view of the general statement in text regarding Pf predominance and author’s response.

Response 5:

i) The requested information is to our knowledge not available. Therefore, we want to keep our current response.

Response 6:

ii) Thus, no need to address this point further.

Methodology :

i) What is the reason and logic of recording microscopy positive results as the geometric means? Pl explain.

ii) What were the inclusion and exclusion criteria of the study participants? How many health facilities were included in the study and what was the no. of study participants in each health facility,

iii) Was there a specific reason for recruiting only < 5 yrs. Children in the study?

Point 7: i) Addressed by authors and accepted.

Point 8: ii) Addressed partially.

The breakup of 391 children has been given in the reply but I couldn’t see it in the revised paper. Pl add this.

Point 9: iii) Response provided by authors is accepted.

Response 7:

i) No need to address this point further as we have adequately addressed this issue.

Response 8:

ii) We have now added the following information to the result section 3.1: The recruitment according to the sites was as follows: 115 were included at Temnaoré health facility, 91 at Soaw health facility, 82 at Nazoanga health facility, 68 at Paediatric ward of CMA reference hospital, and 35 at Pella health facility (see lines 153-156).

Response 9:

iii) No need to address this point further as we have adequately addressed this issue.

Results & Discussion :

i) The study appears to be biased. How could 100% microscopy results be mono infections with Pf?

ii) Why did the authors not record previous anti-malaria treatment and take this into account when including a patient in the study, given the fact that antigen persistence is expected for at least three weeks after treatment?

iii) So many possibilities for low specificity have been enumerated. Since this was a controlled study specifically designed to assess the diagnostic performance of a particular type of RDT, in such a study the RDTs used are not expected to already deteriorate due to various reasons.

iv) Overdiagnosis of PF infections due to the persistence of antigen and repeated testing with RDTs is well known, but no program is expected to procure and use RDTs with specificity as low as 40 %.

v) What is the message of the RDT policy of the country? How do the authors perceive this study to be useful for the Program?

Point 10: i) Addressed; response   accepted.

Point 11: ii) Not addressed. However, since the authors do not have a way of addressing this issue now, the response of authors is accepted.

Point 12: iii) It seems the authors have not been able to understand the query. This is related to the quality assurance mechanisms of RDTs as per WHO guidelines. The RDTs procured and used by the NMCP are expected to be subjected to quality checks through a quality assurance system. In case such mechanisms are in place , it is desirable to add the same under the quality assurance/quality control heading as already explained above. This will add value to the paper.

Point 13: iv) Again, this is related to the quality assurance of RDTs. If the system exists , pl add under relevant heading as already explained above.

Point 14: v) Though the authors have responded to this, it is advised that in view of above comments and depending upon the quality assurance mechanisms present in the country under the NMCP, the appropriate information should be provided under the quality assurance heading. In case, no such mechanisms exist, this should be written clearly and accordingly

Response 10:

i) No need to address this point further as we have adequately addressed this issue

Response 11:

ii) Thank you.

Response 12:

iii) To our knowledge, the RDTs procured and used by the NMCP of Burkina Faso are not subjected to a quality assurance system. The quality of their performance is based on the internal quality control of the RDTs products procured by the manufacturers and followed by the health workers who use them to consider if the results are right or not in according to the control line present or not on the RDT cassettes during the malaria testing performed. We now added this information to the quality control section to clarify it (see 2.4. quality control: lines 135-136).

Response 13:

iv) See comment above

Response 14:

v) See comment above

We hope that we have now sufficiently addressed the issues raised by the three reviewers and that you will consider our manuscript for publication in your journal.

Yours sincerely,

On behalf of all co-authors

Massa dit Achille Bonko.
